# Assembly and functionality of the ribosome with tethered subunits

Nikolay A. Aleksashin [1], Margus Leppik[2], Adam J. Hockenberry [3,4], Dorota Klepacki[1], Nora Vázquez-Laslop [1], Michael C. Jewett [3], Jaanus Remme[2] & Alexander S. Mankin [1]

Ribo-T is an engineered ribosome whose small and large subunits are tethered together by linking 16S rRNA and 23S rRNA in a single molecule. Although Ribo-T can support cell proliferation in the absence of wild type ribosomes, Ribo-T cells grow slower than those with wild type ribosomes. Here, we show that cell growth defect is likely explained primarily by slow Ribo-T assembly rather than its imperfect functionality. Ribo-T maturation is stalled at a late assembly stage. Several post-transcriptional rRNA modifications and some ribosomal proteins are underrepresented in the accumulated assembly intermediates and rRNA ends are incompletely trimmed. Ribosome profiling of Ribo-T cells shows no defects in translation elongation but reveals somewhat higher occupancy by Ribo-T of the start codons and to a lesser extent stop codons, suggesting that subunit tethering mildly affects the initiation and termination stages of translation. Understanding limitations of Ribo-T system offers ways for its future development.

[1] Center for Biomolecular Sciences, University of Illinois at Chicago, Chicago, IL 60607, USA. [2] Institute of Molecular and Cell Biology, University of Tartu, Riia 23, 51010 Tartu, Estonia. [3] Department of Chemical and Biological Engineering and Center for Synthetic Biology, Northwestern University, 2145 Sheridan Road, Evanston, IL 60208, USA. [4]Present address: Department of Integrative Biology, Institute for Cellular and Molecular Biology, The University of Texas at Austin, 2500 Speedway, Austin, TX 78712, USA. These authors contributed equally: Nikolay A. Aleksashin, Margus Leppik, Adam J. Hockenberry. Correspondence and requests for materials should be addressed to J.R. (email: jaanus.remme@ut.ee) or to A.S.M. (email: shura@uic.edu)

Ribosomes consist of two subunits, small and large (30S and 50S in bacteria). The small subunit interprets the genetic information by selecting aminoacyl-tRNAs cognate to the mRNA codons in the decoding center. The large subunit carries the catalytic peptidyl transferase center (PTC) where amino acids are polymerized into a protein. Small and large subunits unite together at the start codon of a gene to form the 70S ribosome and dissociate again at the stop codon upon completing the synthesis of the encoded protein. The complementarity of the 3′ terminal segment of the 16S rRNA of the small subunit to the Shine–Dalgarno (SD) sequence upstream from the mRNA start codon facilitates the initiation of translation[1]. By altering the SD sequence in an mRNA and introducing a cognate anti-SD (ASD) sequence in the 16S rRNA, it is possible to re-direct a sub-population of small ribosomal subunits to initiate translation of one specific mRNA species, rendering its expression independent of (orthogonal to) the other cellular genes[2,3]. While making an orthogonal 30S subunit is fairly straightforward, engineering a fully orthogonal ribosome—in which both subunits are dedicated to the production of one specific protein—is a daunting task because of the stochastic nature of the small and large subunit association. And yet, constructing truly orthogonal translation machinery is highly desirable because it could carry out new functions that are beyond the reach of the wild type (wt) ribosome. These tasks could range from incorporating non-canonical amino acids into a protein, to synthesizing genetically-programmed non-proteinaceous polymers and many others[4,5]. The fully-orthogonal translation system could also be a useful tool for addressing fundamental questions about the origin, evolution, and functions of the wild type ribosome.

An important step towards engineering of a fully-orthogonal translation system was our construction of Ribo-T, a ribosome with tethered subunits[6]. A similar, but less optimized, design was also proposed by another group[7], which was later improved[8]. In Ribo-T, the circularly permuted (cp) 23S rRNA (which constitutes the major RNA scaffold of the 50S subunit) is grafted into the 16S rRNA (the rRNA component of the 30S subunit), thereby generating a hybrid 16S–23S rRNA molecule with the 30S and 50S subunit rRNA moieties connected via two RNA linkers (Fig. 1a, b). In spite of its peculiar design, Ribo-T with an altered ASD (oRibo-T) is capable of functioning as an orthogonal translation machine in the cell which could be evolved for new functions[6,8]. Furthermore, Ribo-T with the wt ASD sequence in its 16S rRNA segment is sufficiently active to support the growth of cells that completely lack wt ribosomes (Ribo-T cells). However, the evolved Ribo-T cells grow at half the rate of the cells with wt ribosomes, which suggests some limitations of the tethered ribosome[6]. Discernibly, the lethargic growth of the Ribo-T cells reflects some specific limitations of the tethered ribosome system. Understanding what those impediments are is important not only for further improvements of the Ribo-T design but also for engineering new variants of the rRNA-based translation machines. We hypothesized that two main limitations could account for the reduced activity of Ribo-T: aberrant biogenesis and/or limited functionality.

Every cell in a rapidly growing *Escherichia coli* culture needs to put together approximately 70,000 ribosomes during the doubling period[9] and it has been demonstrated that there is a direct correlation between the rate of ribosome assembly and cell growth[10,11]. Ribosome biogenesis includes multiple delicately coordinated and intricately intertwined steps including transcription of rRNA operons (composed of the 16S, 23S, and 5S rRNA genes) and the initial co-transcriptional folding of the primary rRNA transcript, multi-step nucleolytic processing and chemical modification of specific rRNA residues, production of ribosomal proteins (r-proteins) that, in some cases, contain posttranslational modifications, ordered step-wise association of the primary transcript with r-proteins, and final adjustment of the ribosomal subunit structure into the functionally-active state (reviewed in ref. [12]). The intricate process of ribosome biogenesis is assisted by a number of protein factors, many of them with poorly understood functions[12–14].

The permuted arrangement of rRNA sequences in Ribo-T may alter its ability to rapidly assemble into a complex macro-molecular machine. The general order of transcription of the Ribo-T rRNA parts significantly deviates from the transcription of the wt rRNA operon, mostly because the 3′ segment of the 16S rRNA cannot be synthesized until transcription of the entire cp23S rRNA sequence is completed (Fig. 1a, b). Furthermore, the order of synthesis of the cp23S rRNA sequence within the Ribo-T RNA structure is different from that of the wt 23S rRNA (Fig. 1a, b). The altered path of the cp23S rRNA transcription may impact important steps of large subunit biogenesis, disrupting rRNA folding, the introduction of posttranscriptional modifications (PTMs) or the incorporation of r-proteins. Ribo-T maturation could be additionally affected by an imbalanced production of r-proteins and assembly factors due to the altered functionality of the translation apparatus in the Ribo-T cells.

Reductions in functional activity could also lead to the observed diminished rate of protein synthesis and slow growth of the Ribo-T cells. Every step of translation, from initiation, which is normally driven by interactions of the isolated small subunit with mRNA and tRNA, to elongation, which involves inter-subunit rotation, and to termination and recycling, which require splitting of the 70S ribosome, could be negatively affected in Ribo-T. However, the possibly altered functionality of the tethered ribosome at any of these steps has yet to be studied.

In order to find out which specific aspects of biogenesis or functioning of the tethered ribosome might account for the slow growth of Ribo-T cells, we examined Ribo-T assembly and analyzed its protein synthesis capabilities in vivo. We found that the Ribo-T assembly is notably slowed compared to wt ribosomes and likely plays the dominant role in impeding the growth of the Ribo-T cells. Our results illuminate further ways for Ribo-T optimization and underscore opportunities for ribosome engineering.

## Results

**The assembly of Ribo-T is slow.** We examined the course of Ribo-T assembly by pulse-labeling cells with [3H]-uridine, abolishing new rounds of transcription by the addition of rifampicin and following incorporation of radioactivity into Ribo-T particles by sucrose gradient centrifugation. As expected, in the control cells, wt ribosomes assemble rapidly: even at the first time point (5 min after addition of [3H]-uridine to the cells), a significant fraction of radioactivity was already associated with the 30S and 50S subunits or with the 70S ribosomes, and by 15 min most of the radioactivity was in the 70S ribosome peak (Fig. 1c). In contrast, essentially no newly transcribed rRNA could be found in the Ribo-T 70S peak at the 5 or 15 min time points, but instead radioactivity was found in the particles sedimenting at ~55S. Radioactive rRNA could be first detected in the Ribo-T 70S peak 35 min after addition of [3H]-uridine to the culture, and more radioactivity shifted from the 55S material into the 70S peak after additional 30 min. However, unlike wt ribosomes, a substantial amount of [3H]-uridine remained associated with the 55S–60S fraction or was found at the top of the gradient (Fig. 1c) even at the 65 min time point. Besides the 55S peak, no other intermediates with distinct sedimentation characteristics were observed in the Ribo-T cells, suggesting the existence of a single assembly bottleneck.

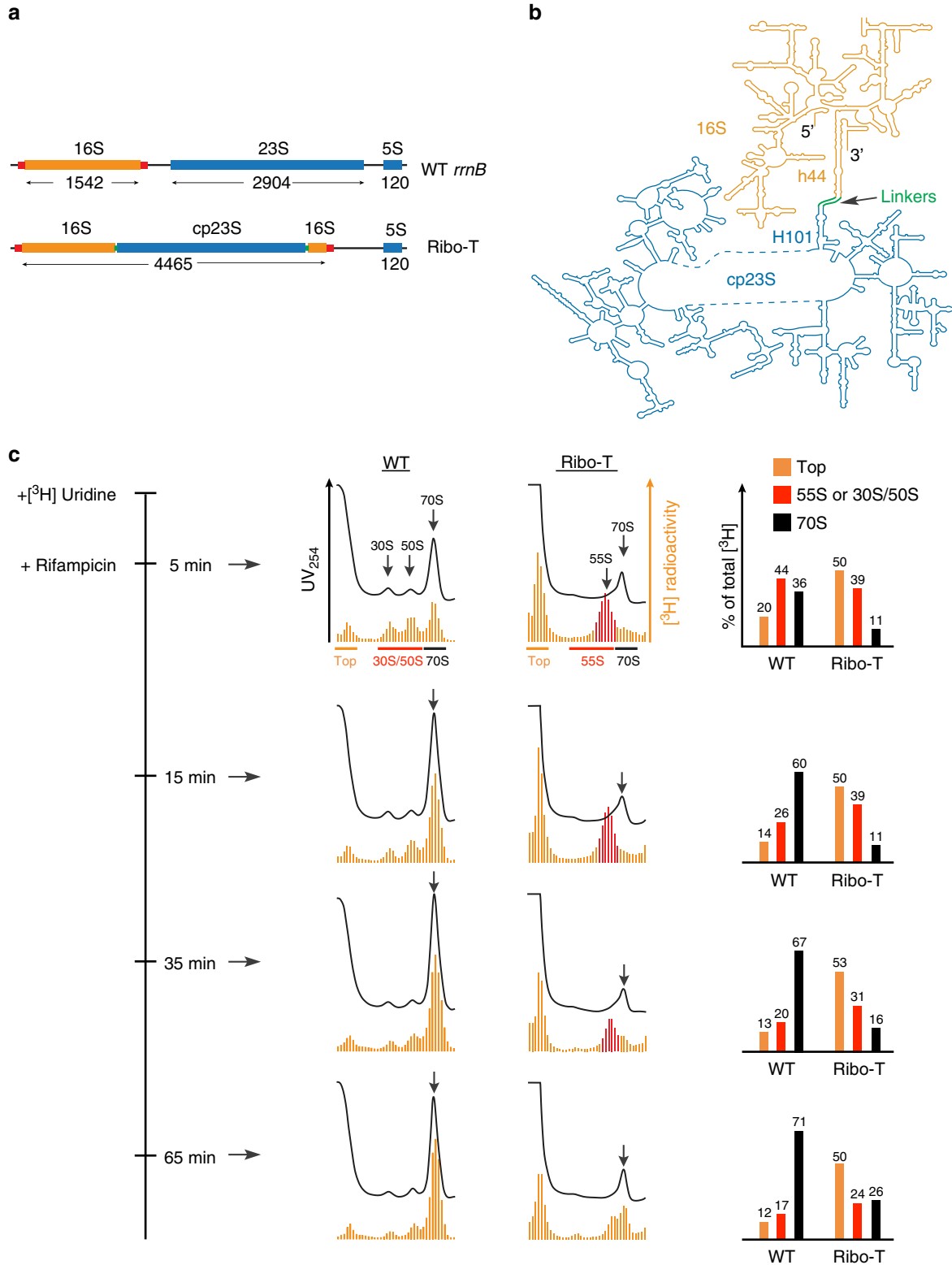

Taken together, our results showed that the Ribo-T biogenesis is significantly impeded compared to the wt ribosome primarily due to a slow conversion of the 55S assembly intermediates into mature 70S Ribo-T particles and possible formation of the dead-end assembly products that eventually degrade without being converted into functional Ribo-T. Thus, imperfect Ribo-T maturation is one of the factors that likely curtail its ability to support fast cell growth.

**Distinct Ribo-T rRNA residues remain under-modified**. PTMs play an important role in the biogenesis and function of the ribosome[15,16]. Introduction of PTMs in rRNA could serve as checkpoints to ensure the appropriate sequence of ribosome assembly steps[17,18]. Because our pulse-labeling experiments indicated that Ribo-T biogenesis is stalled at a specific stage of the assembly and because Ribo-T functionality is diminished compared to wt ribosomes[6], we wondered whether the required rRNA

**Fig. 1** The Ribo-T biogenesis is stalled at a specific stage of assembly. **a** The structures of the *E. coli* wt rRNA operon and Ribo-T (tRNA genes in the 16S/ 23S rRNA spacer are not shown). The 16S rRNA flanking segments that form the processing hairpin are indicated with red bars. Length of each rRNA in nt is indicated. **b** The schematics of the Ribo-T rRNA shown in the secondary structure diagram depicting the insertion of circularly-permuted 23S rRNA (cp23S) into loop of helix 44 (h44) of the 16S rRNA. The linkers connecting h44 in the 16S rRNA to H101 in the 23S rRNA are indicated. The mature 5′ and 3′ and of the Ribo-T rRNA are marked. **c** Sucrose gradient analysis of Ribo-T and wt ribosomes prepared from cells pulse-labeled with [$^3$H] uridine and then treated with rifampicin. The time course of the experiment is indicated on the left. Black traces: the UV$_{254}$ absorbance (the UV peaks representing mature 70S ribosomes are marked by arrows). Orange bars—radioactivity (the 55S Ribo-T assembly intermediates are indicated as red bars). Bar graphs on the right indicate the relative amounts of radioactivity (% of the total radioactivity) on the top of the gradients (orange), in Ribo-T 55S assembly intermediates or 30S/50S subunits (red) or in 70S ribosomes (black). The fractions used for calculation of the corresponding radioactivity values are indicated at the "5 min" gradients. Source data for panel **c** can be found in the Source Data file

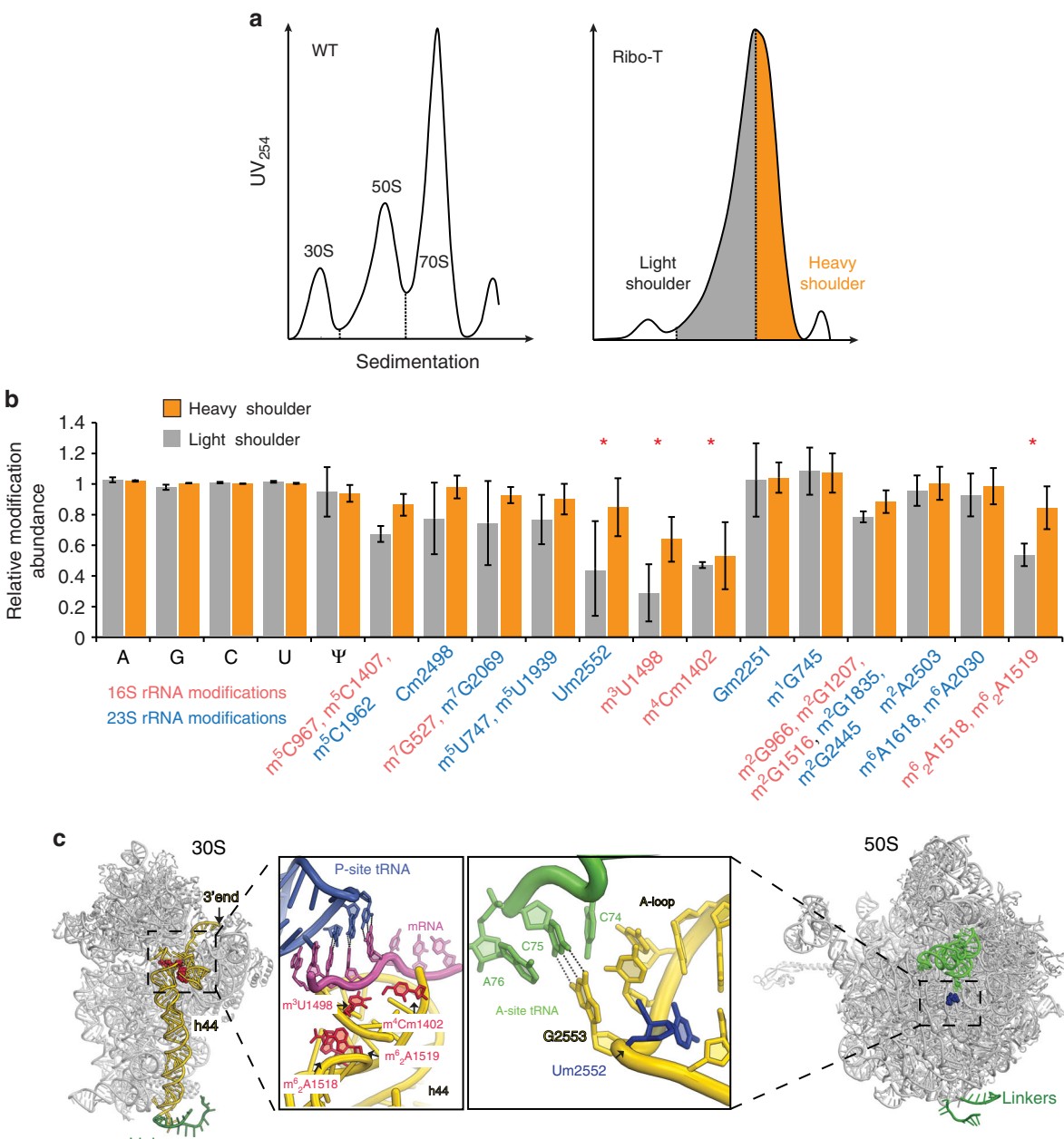

**Fig. 2** Specific posttranscriptional modifications are underrepresented in Ribo-T assembly intermediates. **a** Sedimentation profiles (A$_{254}$) of wt ribosomes and Ribo-T. The light shoulder (gray) and heavy shoulder (orange) material of the Ribo-T peak were collected and analyzed separately. **b** The presence of PTMs in Ribo-T rRNA relative to the wt rRNA. The small subunit PTMs are marked in red, the large subunit PTMs are shown in blue. PTMs present in the LS material at a ≤0.6 level in comparison with the mature wt ribosomes are indicated with the asterisks. Error bars represent deviation from the mean (*n* = 4). **c** The location of the under-modified residues in the small (left) and large (right) subunits of Ribo-T. P-site tRNA (P-tRNA) is blue, A-site tRNA (A-tRNA) is green, and mRNA is colored magenta. Source data for panel **b** are provided in the Source Data file

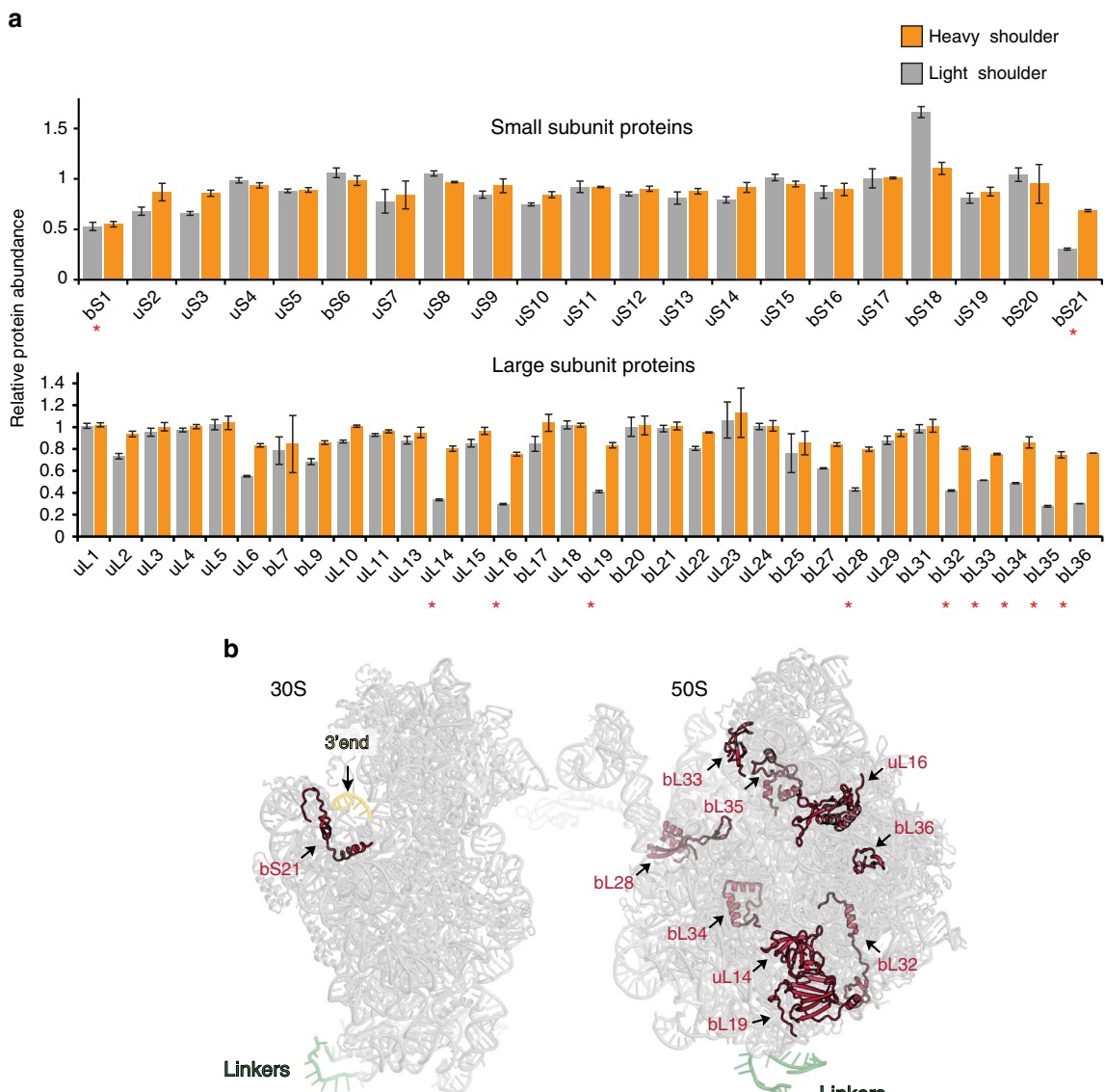

**Fig. 3** Specific ribosomal proteins are lacking in the Ribo-T precursor. **a** The results of the quantitative proteomics analysis of the presence of individual r-proteins in mature Ribo-T (the heavy shoulder material) and assembly intermediates (the light shoulder material) from the Ribo-T peak. Red asterisks indicate those r-proteins that are reproducibly underrepresented in the Ribo-T assembly intermediates (less than 60% of the wt 70S ribosome control). Source data are provided in the Source Data file. Error bars represent deviation from the mean ($n = 2$). **b** Location of the underrepresented proteins (red) in the small and large ribosomal subunits

PTMs are properly represented in the precursors and in the mature Ribo-T.

Although no distinct 55S peak could be observed in the sucrose gradient UV profile of unlabeled Ribo-T (Fig. 1c), indicating that the steady-state level of the Ribo-T assembly intermediates is not very high, the Ribo-T UV peak in the preparative isolation experiment was asymmetric exhibiting the presence of a trailing light shoulder (LS) (Fig. 2a). We surmised that the leading shoulder of the 70S peak contains the fully-assembled Ribo-T whereas the trailing shoulder is likely enriched in the stalled assembly intermediates. Therefore, for the subsequent experiments we arbitrarily separated the Ribo-T peak into the LS and heavy shoulder (HS) fractions (Fig. 2a) and individually analyzed their PTM content by HPLC[17] (see Source Data file).

While all the known ribosome PTMs amenable to our analysis could be detected in Ribo-T rRNA, some of them were present in a reduced amount in the LS material that is presumably enriched with immature Ribo-T precursors (Fig. 2b). In particular, four of the 16S rRNA PTMs ($m^3U1498$, $m^4Cm1402$, and $m^6_2A1518$/$m^6_2A1519$) and one 23S rRNA modification (Um2552) (Fig. 2c) were notably underrepresented in the Ribo-T assembly intermediates (marked by asterisks in Fig. 2b). In the HS material, corresponding to mature Ribo-T, dimethylation of A1518/A1519 in the 16S rRNA and modification of Um2552 in the 23S rRNA were notably increased, whereas the modification of two 16S nucleotides, U1498 and C1402, still remained below 70% (Fig. 2b). The individual absence of any of the latter 16S rRNA modifications does not substantially affect fitness of wt E. coli cells[19,20] and by extrapolation, the lack of these PTMs is not expected to importantly impair Ribo-T functionality.

While underrepresentation of particular 16S rRNA PTMs is not expected to be the major culprit of the slow growth of Ribo-T cells, the incomplete 2′-methylation of U2552 in the 23S rRNA, which is present at 40% in the Ribo-T LS material (Fig. 2b), could be an important factor. It has been shown that E. coli cells deficient in the activity of the methyltransferase RlmE responsible

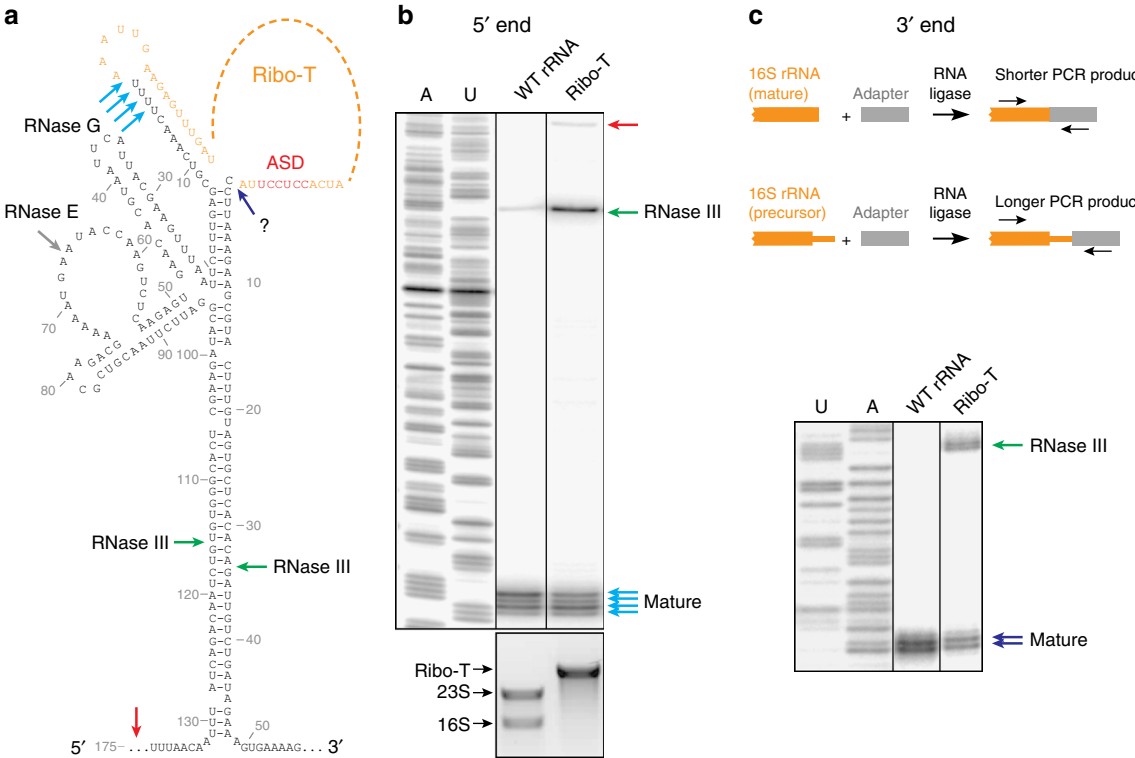

**Fig. 4** Processing of the Ribo-T rRNA ends is delayed. **a** Putative secondary structure of the 16S rRNA (and Ribo-T rRNA) flanking regions[62]. The rRNA processing cleavage sites and the corresponding known nucleases are indicated. The nuclease responsible for the final steps of the 3′ end processing is unknown (a question mark). The wt ASD is indicated in red. **b** Analysis of the 5′ end processing by primer extension. The primer extension product representing rRNA template generated by RNase III cleavage is indicated with the green arrow. Red arrow indicates the cDNA product generated on the template that remains uncleaved by RNase III. The U- and A-specific sequencing reaction were generated using pAM552 plasmid carrying *rrnB* operon[6] as a template. The agarose gel at the bottom shows the long rRNA species present in the cells from which the corresponding rRNA was prepared for the analysis. **c** Top: the scheme of the 3′ RACE experiment. Bottom panel: analysis of the 3′ RACE PCR products in the sequencing gel. The DNA bands representing mature rRNA and the rRNA species generated by the RNase III cleavage are indicated by the blue and green arrows, respectively. The U- and A-specific sequencing reactions, used as a ruler, were generated using pAM552 plasmid[6] as a template. Source data are provided in the Source Data file

for this modification accumulate assembly intermediates of the large ribosomal subunit lacking several r-proteins[21,22]. Therefore, in order to explore whether incomplete modification of U2552 in Ribo-T intermediates complicates the association of particular r-proteins impeding late maturation steps, we analyzed the protein composition of the material in the LS and HS fractions of the Ribo-T peak.

**Some proteins are underrepresented in Ribo-T assembly intermediates**. The protein content of the LS and HS components of the Ribo-T peak (Fig. 2a) was analyzed by mass-spectrometry (Fig. 3a). Two small subunit proteins, bS1 and bS21, and nine large subunit proteins, uL14, uL16, bL19, bL28, bL32–bL36 (marked by asterisks in Fig. 3a), were present in less than half of their stoichiometric amounts in the LS fraction enriched in assembly intermediates. Additionally, both of the 30S proteins (bS1 and bS21) remained notably underrepresented in the mature Ribo-T (HS material) (Fig. 3a). Protein bS1 is known to be loosely bound to the 30S subunit[23] and could be easily lost during Ribo-T isolation. Protein bS21, which is one of the late assembly proteins[24,25], binds in the vicinity of the 3′ minor domain of the 16S rRNA, whose accommodation into the Ribo-T structure could be delayed (Fig. 3b). However, the bS21-encoding gene *rpsU* is not essential in *E. coli*[26] and the paucity of this protein in the mature Ribo-T is not expected to have a dramatic effect on its activity.

A subset of the 50S proteins lacking in the Ribo-T precursor (uL16, bL19, bL28, bL35, and bL36) (Fig. 3b) were previously found to be underrepresented or even completely lacking (bL36) in the assembly intermediates accumulated when the U2552 methylation was abolished in wt *E. coli* cells[21,22]. Thus, the sluggish modification of U2552 during Ribo-T assembly could be the key factor that delays the incorporation of these proteins, resulting in the accumulation of the assembly intermediates. In agreement with this notion, the protein content of the large subunit was nearly completely restored in the mature Ribo-T (Fig. 3a).

**Delayed processing of Ribo-T rRNA ends**. Nucleolytic processing of the rRNA primary transcript and generation of the mature rRNA ends are important steps of the ribosome biogenesis required for the formation of functionally-active ribosomal particles[27,28]. Therefore, we asked whether the unusual architecture of Ribo-T and the detected problems in its biogenesis may affect the maturation of the rRNA ends.

For these experiments, we analyzed the total rRNA extracted from exponentially growing wt or Ribo-T cells. When the extent of the 5′ end processing was interrogated by primer extension, we noted a long extension product in the Ribo-T sample that was absent in wt (red arrowhead in Fig. 4a, b, see also Supplementary Fig. 1a). This product represents the precursor extending beyond the RNase III cut sites (green arrowheads in Fig. 4a), indicating that cleavage by RNase III, which requires the formation of the

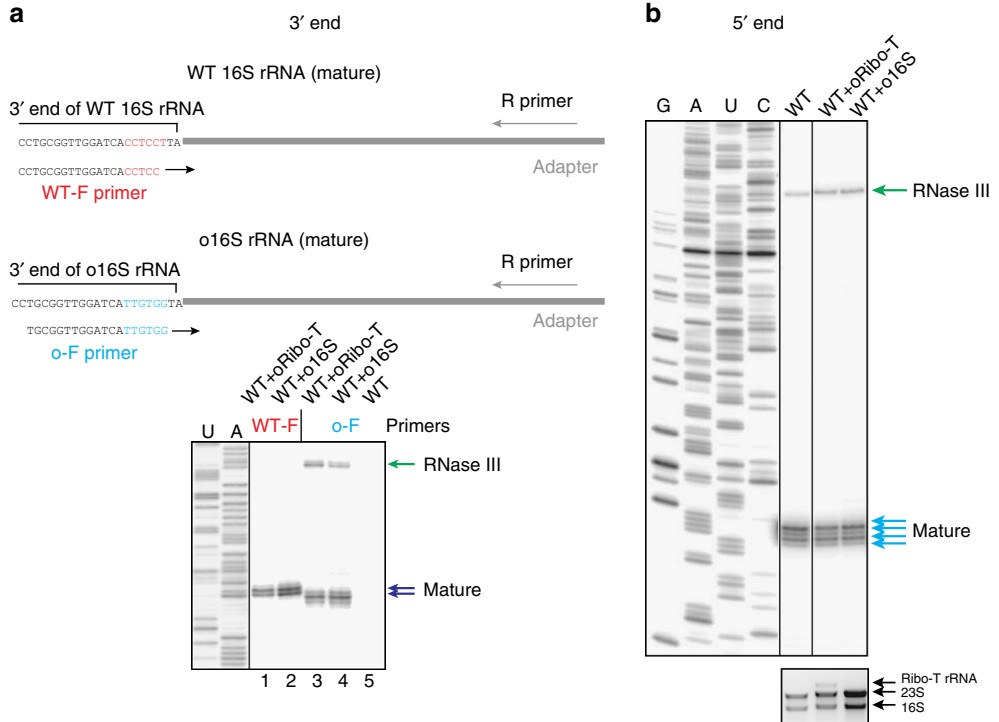

**Fig. 5** The ends of orthogonal Ribo-T and 16S rRNA are properly trimmed. **a** Top: the scheme of the ASD-specific 3′ RACE experiment. Forward PCR primers specific for wt ASD (wt-F) or oASD (o-F) were used at the PCR step of the 3′ RACE procedure to distinguish the processing of the corresponding rRNA species. The wt SD sequence is shown in red and orthogonal SD sequence is shown in blue. Bottom: Analysis of the 3′ RACE PCR products in the sequencing gel. Note that all the cells used for the preparation of rRNA contained wt ribosomes in addition to oRibo-T or o16S rRNA. PCR products generated using the wt-F/R primer combination reveal the processing of wt 16S rRNA; PCR products generated using the o-F/R primer combination reveal the processing of oRibo-T or o16S rRNA. Lanes 1 and 2: analysis of wt 16S rRNA processing; lanes 3 and 4: analysis of oRibo-T or o16S rRNA processing. The lack of the DNA bands in lane 5, where the sample did not contain orthogonal rRNA, confirms specificity of the o-F PCR primer for the o-rRNA. The difference in migration of the PCR products generated on the wt or orthogonal templates is due to the different lengths of the WT-F and o-F primers. **b** Analysis of the 5′ end processing of a mixture of wt and orthogonal rRNA by primer extension. Note, that the DNA primer used to prime cDNA synthesis does not discriminate between wt and orthogonal rRNA species (see Supplementary Fig. 1a). In **b**, the agarose gel representing long RNA species is shown below the sequencing gel. Source data are provided in the Source Data file

processing step, is more sluggish in the case of Ribo-T. Noteworthy, in spite of the presence of a longer extension product, the cDNA band representing the RNase III-generated precursor (green arrowhead in Fig. 4b) was more pronounced in the Ribo-T sample compared to the wt control (28% vs. 6%). Accumulation of this precursor suggests that the processing steps subsequent to the RNase III cleavage (involving the activity of RNases E and G) (gray and cyan arrows in Fig. 4a), could be more problematic during Ribo-T rRNA maturation. Qualitatively similar results were obtained when these experiments were performed on RNA extracted from the LS and HS of the Ribo-T peak (Fig. 2a), except that the precursor-enriched LS material contained more 5′-unprocessed rRNA in comparison with mature Ribo-T from the HS fraction (Supplementary Fig. 1).

We next compared the processing of the 3′ end of wt 16S rRNA and Ribo-T by 3′ RACE[29]. Following ligation of an adapter rRNA to the 3′ end of 16S rRNA and performing PCR reactions with 16S rRNA- and adapter-specific primers, the sizes of the resulting DNA were analyzed in a denaturing gel (Fig. 4c, see also Supplementary Fig. 1a) and verified by Sanger sequencing. In contrast to the wt control, where only short PCR products corresponding to the mature 16S rRNA end was detected, the intensity of the DNA bands representing an RNase III-generated processing intermediate was comparable to that of the band corresponding to the mature Ribo-T rRNA (Fig. 4c, green and blue arrows, respectively). Thus, similar to the 5′ end processing, the final trimming of the Ribo-T rRNA 3′ terminus is less efficient

in comparison with wt 16S rRNA and a significant fraction of Ribo-T rRNA carries a 3′-extension.

In conclusion, investigation of the Ribo-T biogenesis showed the existence of an assembly bottleneck resulting in a slow conversion of precursors into mature Ribo-T. Thus, inefficient assembly emerged as one important factor that could be responsible for the slow growth of the Ribo-T cells.

**Altering ASD does not change the final rRNA processing point**. Because one of the possible Ribo-T applications is its use as an orthogonal translation system[6], we also tested how altering the ASD sequence would affect the maturation of the Ribo-T rRNA 3′ end. The wt ASD CCUCCU (shown in red in Figs. 4a and 5a) was changed to an oASD sequence UUGUGG (shown in blue in Fig. 5a) and the resulting orthogonal Ribo-T (oRibo-T) was expressed in the *E. coli* strain SQ171[30] that maintains expression of wt ribosomes. As a control, we also expressed in the same host the dissociable ribosome carrying the same oASD in its 16S rRNA. By using wt- and oASD-specific forward PCR primers, we were able to distinguish and directly compare processing of the 3′ ends of wt 16S rRNA and oRibo-T rRNA (Fig. 5a). Changing the ASD sequence did not alter the final processing point and as a result mature o16S rRNA or mature oRibo-T rRNA ended with the adenine residue corresponding to the 3′-terminal A1542 of the wt 16S rRNA. While altering 16S rRNA ASD sequence compromised the 3′ end processing (Fig. 5a, lane 4) it did not

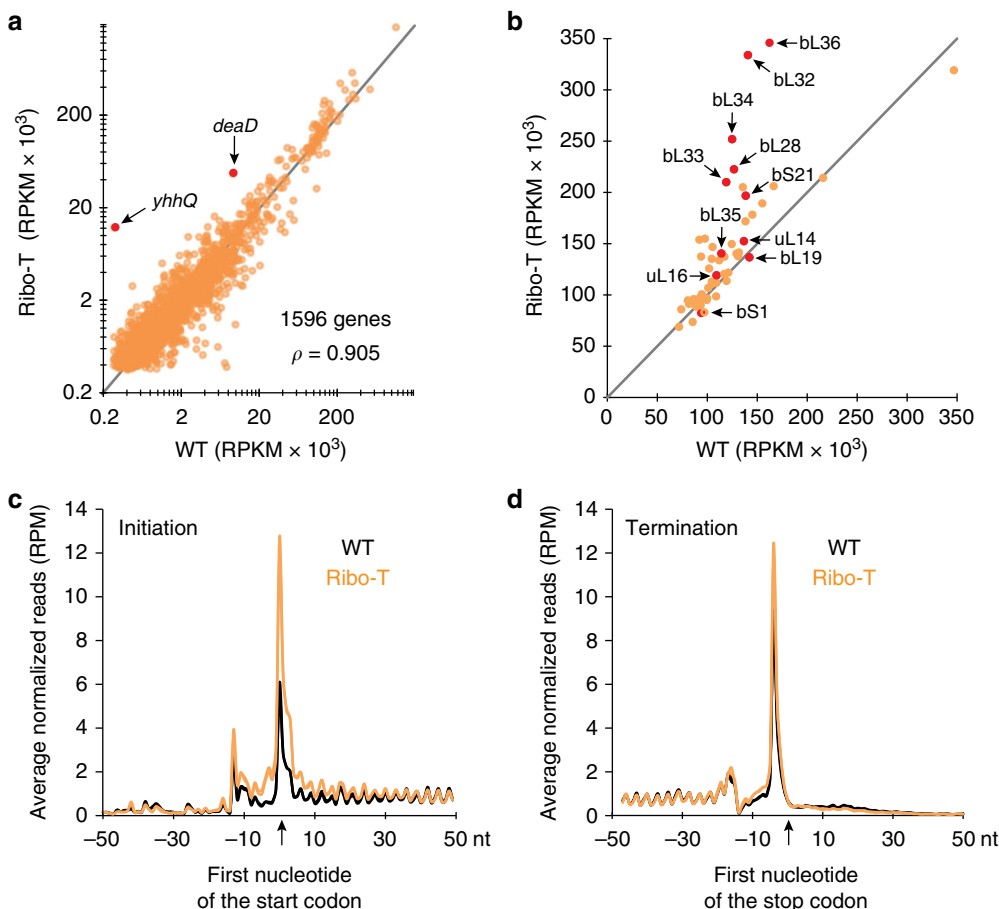

**Fig. 6** Ribo-seq analysis of translation in the Ribo-T cells. **a** Expression of the genes in the Ribo-T and wt cells calculated via average RPKM (reads per kilobase per million mapped reads) values across the biological replicates ($n = 2$) for Ribo-seq data. The dots representing the *yhhQ* and *deaD* genes are highlighted. **b** Expression of r-proteins in cells expressing Ribo-T or wt ribosomes calculated from Ribo-seq data as in panel **a**. Red dots indicate proteins that are underrepresented in the stalled Ribo-T assembly intermediates. Note that many of the underrepresented proteins are apparently overexpressed in the Ribo-T cells, indicating that their diminished presence in the intermediates is not caused by a low level of expression. **c, d** Metagene analysis illustrating the difference in the **c** start- and **d** stop-codon occupancy in the Ribo-T and wt cells

further exacerbate the already slow 3′ end processing of Ribo-T (compare Figs. 4c and 5a, lane 3). Altering the ASD sequence had negligible effect on the processing of the rRNA 5′ end (Fig. 5b). Thus, rRNA maturation in oRibo-T does not present additional problems when compared to Ribo-T constructs containing the wt ASD sequence.

**Ribo-T can efficiently translate *E. coli* genes**. In spite of the stalled assembly, a sufficient amount of functionally-active mature Ribo-T is generated in the cell to support cell growth. Nevertheless, the functional properties of the fully-assembled Ribo-T may differ from those of wt ribosomes. This can be due to the incomplete PTM of some of the rRNA residues (Fig. 2b), the incomplete incorporation of some ribosomal proteins (Fig. 3), or the incomplete trimming of the Ribo-T 16S rRNA ends (Fig. 4). However, the main factor that could complicate Ribo-T functionality is the presence of the artificial intersubunit linkers, because they could restrict the rotational dynamics of the subunits.

In order to assess the Ribo-T functionality in the living cell, we employed ribosome profiling (Ribo-seq)[31–33] to examine translation in the exponentially growing *E. coli* cells expressing either Ribo-T or wt ribosomes. In parallel, we also collected RNA-seq

data that made it possible to control for changes in the cellular transcriptome and to test whether any genes showed differential translation efficiency in cells expressing Ribo-T compared to dissociable ribosomes. The Ribo-seq and RNA-seq experiments were carried out in two biological replicates that generated highly converging results (Supplementary Fig. 2). After filtering the reads mapping to the stable RNA genes, 14–26 million reads were mapped to the *E. coli* (MG1655) genome in the Ribo-seq datasets, whereas RNA-seq experiments generated 5.1–7.6 million mapped reads (Supplementary Table 1).

We used the Ribo-seq data to analyze the translation of individual genes (Fig. 6a). While the translation level of the majority of the genes was similar in Ribo-T and wt cells, there were changes in the relative expression of several genes. In Ribo-T cells, 17 genes showed more than a 2-fold up-shift in expression at a statistically-significant level (Student's *t*-test, false discovery rate (FDR) corrected $p < 0.02$) and 38 genes were down-regulated more than 2-fold (Supplementary Fig. 3a and Supplementary Table 2). A number of the up-regulated genes (e.g., *bhsA, uspG, uspF, yefM*) encoded proteins related to stress response, which is not surprising given the slow growth rate of Ribo-T cells. Noteworthy, the products of several up-regulated genes are related to ribosome biogenesis or translation. For example, helicase DeaD involved in ribosome biogenesis is expressed >5

times in Ribo-T cells (Fig. 6a and Supplementary Table 2). Another assembly factor, ObgE, is overexpressed by ~3.3-fold (Supplementary Table 2). This trend is driven by an increased mRNA abundance (likelihood ratio test, FDR corrected $p < 0.001$, Supplementary Fig. 3a,b and Supplementary Data 2) and likely reflects an attempt of the Ribo-T cells to compensate for the somewhat reduced protein synthesis rate caused by the slow ribosome assembly. The most strongly activated gene in Ribo-T cells was *yhhQ* (>40-fold) (Fig. 6a, Supplementary Fig. 3a, Supplementary Table 2 and Supplementary Data 1). Its over-expression was also primarily due to a higher mRNA abundance (likelihood ratio test, FDR corrected $p < 0.001$, see Supplementary Data 2), but also to a slightly higher translation efficiency (Supplementary Fig. 3b,c). The encoded protein, YhhQ, is involved in the biosynthesis of queuosine, a precursor of a PTM found in the anticodons of Tyr, His, Asn, and Asp tRNA species[34,35]. Even though there is an obvious link of YhhQ to translation, the reasons for the overexpression of this gene in the Ribo-T cells remain unknown.

Although only one r-protein gene, *rpmF*, exhibited a robust overexpression in the Ribo-T cells (>2-fold, Student's *t*-test, FDR corrected $p < 0.02$), the production of many of the remaining r-proteins also appeared increased in the Ribo-T cells (Fig. 6b, Supplementary Fig. 3a and Supplementary Data 1). Importantly, the list of the r-protein genes with apparently increased Ribo-seq reads includes a number of those that were underrepresented in the Ribo-T precursors (Fig. 3), indicating that their poor incorporation is not due to inefficient expression of their genes.

The Ribo-Seq data allowed us to gain insights into the particularities of mRNA translation by Ribo-T. Metagene analysis showed a ~2.1-fold increase in the average ribosome density at the start codons of the genes (Fig. 6c) (mixed linear-model, $p < 10^{-16}$). We also noted a slightly increased density of footprints (~30%) near the 3′ ends of the genes (Fig. 6d), indicating that tethering of the ribosomal subunits slightly affects not only initiation but possibly the termination/recycling steps of translation as well.

Differences in Ribo-seq data between Ribo-T and wt cells could be a result of either changes in mRNA abundances or altered translation efficiencies of individual mRNAs. However, after FDR correction, none of the 1596 genes with adequate read coverage in all Ribo-seq datasets displayed significantly different translation efficiency between Ribo-T and wt cells (Supplementary Fig. 3c). Thus, changes in Ribo-seq data for individual genes (Fig. 6a) are largely driven by changes in mRNA abundances. Notably, our Ribo-seq results did not reveal any major defects in translation elongation, such as stalling of Ribo-T at specific sites within the genes or increased frequency of abortive translation. Thus, within the limits of our analysis, we conclude that Ribo-T can efficiently progress through the protein-coding sequence, implying that linking subunits with the RNA tethers does not impair the ability of the ribosome to select aminoacyl-tRNAs, form peptide bonds, and translocate along mRNA. Altogether, it appears that once assembled, Ribo-T can quite efficiently translate all the *E. coli* genes.

## Discussion

It is striking that Ribo-T, whose rRNA architecture is so different from that of the wt ribosome, can be assembled in the cell into a functionally-active macromolecular complex capable of inter-preting the genetic information and synthesizing proteins. Apparently, the framework of one of the most ancient and critical cellular machines is malleable enough to allow successful assembly and function in spite of the dramatic perturbations of its structure. Nevertheless, it was essentially inevitable that the first

rendition of the ribosome with tethered subunits would yield a prototype that would need further improvement. Identifying the key Ribo-T problems is a critical pre-requisite for its further development. Our results have shown that while Ribo-T activity is only marginally affected by tethering of the small and large ribosomal subunits, its biogenesis is notably slower in comparison with that of the wt ribosome. Ribo-T assembly stalls at a specific stage of its maturation, resulting in a sluggish conversion of the intermediates into the mature form. Several experimental results argue that the blockage of Ribo-T assembly occurs at the fairly late steps of its biogenesis: (i) in the RNA-labeling experiments, the stalled intermediates sediment as well-defined 55S particles, indicating that both 30S and 50S subunits of Ribo-T are already sufficiently compact—a feature that is characteristic of the later stages of small and large subunit assembly; (ii) the intermediates contain near-stoichiometric amounts of most of the ribosomal proteins, with only a limited number of r-proteins being under-represented; (iii) rRNA in the stalled Ribo-T precursors carry most of the natural PTMs found in the mature rRNA, including those which are introduced at the late stages of ribosome assembly. Thus, it is apparent that the initial steps of the Ribo-T biogenesis take place fairly rapidly, but assembly then stalls—possibly due to the difficulty in progressing through some critical checkpoint(s). While our data strongly suggest that the 55S particles are converted into mature Ribo-T, it is possible that a fraction of them represent dead-end assembly products that undergo degradation.

Several PTMs of the rRNA of the small and large subunits are underrepresented in the Ribo-T assembly intermediates. In spite of the underrepresentation of some of the 16S rRNA PTMs, their paucity is unlikely to be a major contributor to the slow growth of the Ribo-T cells. Although the N6-dimethylation of A1518 and A1519 was proposed to serve as a checkpoint for the small sub-unit late assembly[36–38], *E. coli* cells lacking KsgA, the enzyme responsible for this modification, do not show any significant growth defects[39]. Similarly, the lack of the enzymes responsible for modification of U1498 and C1402 is not critical for the cells (at least when grown under laboratory conditions)[19,20]. Remarkably, all the undermodified 16S rRNA nucleotides are clustered in the 3′-terminal domain. Sluggish modification of these rRNA residues is likely explained by the delay in tran-scription and incorporation of the 16S 3′-terminal segment into the Ribo-T structure due to the insertion of the entire cp23S rRNA sequence in the loop of h44 in the 16S rRNA (Fig. 1b). Thus, reduced modification of some 16S rRNA residues is per-haps a consequence of prolonged Ribo-T rRNA folding rather than the cause of the assembly defects. Analysis of modified rRNA residues in the Ribo-T precursors has illuminated impor-tant aspects of installation of PTMs in the wt ribosome. In par-ticular, not only the modification of the nucleotide U1498 in the 3′-proximal strand of h44 whose transcription is temporarily delayed in Ribo-T, but also rRNA residue C1402 that belongs to the 5′ strand of the h44 stem showed reduced modification. This observation suggests that the enzymes RsmH and RsmI respon-sible for C1402 methylation recognize not the linear rRNA sequence but rather the h44 structure within the context of the 30S subunit, a view compatible with the results of previous bio-chemical studies[19].

The incomplete 2′O-methylation of U2552 of the 23S rRNA segment of Ribo-T (Fig. 2b) could be an important factor that slows progression through the late assembly step(s). This residue belongs to the functionally important rRNA element involved in positioning the aminoacyl-tRNA in the PTC A site[40–42] (Fig. 2c). U2552 is modified by the enzyme RlmE, whose absence or inactivation causes a severe growth defect[36] and leads to accu-mulation of the 50S subunit assembly intermediates lacking

specific r-proteins[21,22]. Noteworthy, some of the corresponding r-proteins, (uL16, bL19, bL28, bL35 and bL36) were under-represented in the stalled intermediates of the Ribo-T assembly (Fig. 3). Proximity of the proteins bL19 and bL14 to H101, which in Ribo-T is linked to 16S rRNA, could be an additional factor that complicates their incorporation. The proteins uL16, bL28, bL35, and bL36 are known to be underrepresented in the ribosome large subunit precursors accumulating due to the lack of ribosome biogenesis factors[43,44] or depletion of an essential ribosomal protein bL17[14]. These proteins belong to the late assembly proteins that are likely responsible for the fine-tuning of the structure of the PTC[45].

Because the altered order of transcription of cp23S rRNA has fairly small effect on the 50S subunit biogenesis[46], it is likely that the major factor that influences the assembly of the large Ribo-T subunit is the adjacency of the tethered small subunit that complicates access of the RNA modifying enzymes and assembly chaperons to the large subunit interface where U2552 is located. Since most of the PTMs in rRNA are confined to the 16S and 23S rRNA segments located at the subunits' interface it is possible that the evolutionary preservation of the ribosome as a two-subunit particle has been driven not only by the necessity of maintaining the proper dynamics of translation, but also by the requirement of achieving the proper rRNA modification landscape.

The temporal lag in formation of the 16S rRNA processing stem and the increased spatial separation of the 5′ and 3′ strands of the stem in the Ribo-T primary transcript (Fig. 4a) could be the reason for the less efficient nucleolytic processing of the Ribo-T rRNA ends (Fig. 4b, c). Importantly, altering the sequence of ASD to an orthogonal one had no further negative effect on the maturation of the Ribo-T rRNA 3′ terminus (Fig. 5a) suggesting that rendering Ribo-T orthogonal will not additionally complicate its rRNA processing. While the presence of orthogonal ASD in 16S rRNA of dissociable 30S subunit did slow its 3′ end maturation, it did not alter the sequence coordinates of the 3′ end of the mature 16S rRNA (Fig. 5a). This result shows that the ASD sequence even in wt ribosome, which is universally conserved in bacteria and archaea, does not define the final rRNA trimming point. The essential GTPase Era was proposed to facilitate the 16S rRNA 3′ end processing by directly binding to the wt ASD sequence[47,48]. Our data argue that either interactions of Era with the ribosome do not rely on the ASD sequence, or that such interactions are dispensable for the 16S rRNA 3′-end maturation.

The delayed assembly of Ribo-T emerges as one of the major factors that causes slow growth of the Ribo-T cells and is likely the main impediment to the overall Ribo-T functionality. Understanding of this fact illuminates specific approaches that can be undertaken for Ribo-T improvement. It is conceivable that modulating the interaction between the 5′ and 3′ segments of h44 by altering their sequence could accelerate the rate of incorporation of the 3′-terminal minor domain of 16S rRNA into the Ribo-T structure. The Ribo-T RNA linkers that connect cp23S rRNA to the h44 loop should also play an important role in coordinating small and large subunit assembly. For the sake of simplicity, in the initial Ribo-T design those linkers were composed of oligo(A) stretches. Finding linker designs that would better comply with the assembly requirements could improve Ribo-T properties. It is also possible that the overexpression of some ribosome-specific or general RNA chaperones could increase the efficiency of rRNA folding in the Ribo-T context and thus accelerate the assembly. Although our pilot attempts to facilitate Ribo-T assembly by individually overexpressing some proteins involved in ribosome biogenesis, such as Der, ObgE, DbpA, CsdA, SrmB, KsgA, or RelE have not produced any significant improvement, there are many other candidates to try.

These could include general RNA chaperones, such as Hfq, whose production is already increased by ~2-fold in the Ribo-T cells (Supplementary Data 1), other specific ribosome biogenesis factors, or the subunit anti-association factor IF3 that could help to keep immature Ribo-T subunits apart.

While the biogenesis of Ribo-T seems to be the major problem, the fully assembled tethered ribosomes appear to be highly active in translation. The only prominent effect observed in the meta-gene analysis was the increased relative occupancy of the start codons by Ribo-T in comparison with wt ribosomes (Fig. 6c), which confirmed our previous findings obtained in a cell-free translation system[6]. The Ribo-seq data showed that the Ribo-T dwell time at the start codons of many genes is prolonged. Disengagement of the ribosome from the SD sequence may require certain structural maneuvers[49] that could be more energetically-costly when the subunits are tethered together by the RNA linkers or due to the presence of the 3′ extensions retained in a fraction of Ribo-T. It is also possible that dissociation of the initiation factors that is required for the formation of the 70S initiation complex could be hindered in the case of Ribo-T due to the proximity of the tethered large ribosomal subunit.

Ribo-seq analysis did not reveal any major Ribo-T translation problem subsequent to initiation, although we did notice a moderate increase in ribosomal occupancy at stop codons, possibly indicating a mild problem with either translation termination or recycling. Our analysis of translation efficiency changes in 1596 genes showed no significant differences between Ribo-T and wt cells. Thus, it appears that the tethered nature of Ribo-T affects primarily the steps of translation that require subunit joining (initiation) or separation (termination/recycling) but has little effect on the relative mobility of the subunit at the elongation stage of protein synthesis.

In conclusion, our study of the Ribo-T assembly and functionality has shown that slow assembly seems to be the most critical hindrance for fast cell growth and thus, optimization of the Ribo-T design to facilitate its maturation is an important direction for its further improvement. Once assembled, Ribo-T can efficiently synthesize all the cellular proteins, although the somewhat slow initiation of translation is one aspect of its functionality that may require further improvement. Our findings pave the way for the improvement of Ribo-T and other artificial ribosome designs.

## Methods

**Strains, growing conditions, and plasmids.** *E. coli* strain SQ171fg is a derivative of the strain MG1655 (*ilvG rfb-50 rph-1*) that carries additional mutations (Δ*rrn*GADEHBC *yebX*, *rpsA* A549V)[6,30]. The strain also carries the plasmid ptRNA67 (pACYC *ori*, Spc^r) that supplies missing tRNA genes[30] and in addition, plasmids with various versions of rRNA genes that are listed in Supplementary Table 3.

**Pulse labeling studies of the Ribo-T assembly.** *E. coli* SQ171 cells expressing wt ribosomes or Ribo-T were grown at 37 °C in 200 ml of labeling media (10 g/L tryptone, 1 g/L yeast extract, 10 g/L NaCl). When cultures reached the optical density $A_{600nm} = 0.3$, 16 μCi of [$^3$H]-uridine (Hartmann Analytic, specific activity 42 Ci/mmol) was added. After 5 min incubation, subsequent RNA synthesis was inhibited by the addition of 100 mg of rifampicin. Cells were collected at the indicated time points by centrifugation of 50 ml of the culture (10 min, 5000*g*, 4 °C). Cell pellets were resuspended in the LLP buffer (10 mM Tris–HCl, pH 8.0, 60 mM KCl, 60 mM NH₄Cl, 12 mM MgAc₂, 6 mM β-mercaptoethanol) and lysed using Precellys 24 Bead Mill Homogenizer (Bertin Technologies). Lysates were layered onto 10–25% (w/v) sucrose gradient and centrifuged in the SW28 rotor (Beckman) for 17 h at 20,000 rpm (4 °C). The gradients were fractionated into 40 fractions each with the continuous monitoring of the optical absorption ($A_{254nm}$). An equal volume of the 10% trichloroacetic acid (TCA) was added to each fraction, TCA pellets were collected by filtration throw the glass fiber filters (GF/C, Whatman) and the amount of the retained radioactivity was measured by scintillation counting.

**Preparations of Ribo-T and rRNA**. To prepare the Ribo-T and the rRNA for the PTMs and proteins composition analisys overnight cultures were diluted 1:50 in fresh LB media and grown at the 37 °C with shaking to the mid-exponential phase ($A_{600} = 1$). Cells were harvest by centrifugation at 5000 rpm for 10 min (4 °C) in the JA25.50 rotor (Beckman). Cell pellets were resuspended in the lysis buffer 1 (50 mM Tris–HCl, pH 8.0, 60 mM KCl, 60 mM $NH_4Cl$, 6 mM MgAc$_2$, 6 mM β-mercaptoethanol, 16% sucrose) and lysed using the Precellys 24 Bead Mill Homogenizer (Berlin Technologies) according to the manufacturer protocol. Up to 80 $A_{254nm}$ of the lysate were layered onto 15–30% (w/w) sucrose gradient prepared in the LLP buffer and centrifuged 17 h at 22,100 rpm (4 °C) in the SW28 rotor (Beckman). Sucrose gradients were fractionated, fractions containing ribosomes or Ribo-T LS and HS material were combined, diluted with LLP buffer and ribosomes were pelleted by centrifugation (rotor Ti 45, Beckman), 17 h, 38,500 rpm, 4 °C. Ribosome or Ribo-T pellets were dissolved in LLP buffer and stored at −80 °C. rRNA was isolated by phenol–chloroform extraction and ethanol precipitation.

**Analysis of the rRNA post-transcriptional modifications**. The PTM content of rRNA was analyzed by HPLC[50,51]. The extracted rRNA was digested to nucleosides by treatment with P1 nuclease (Sigma) and alkaline phosphatase Fast AP (Thermo Fisher Scientific) as described in ref. [17]. The HPLC separation of the standard and modified nucleosides was done[17,51].

**LC-MS/MS analysis of ribosomal proteins**. Ribosomal protein content of Ribo-T was determined using quantitative mass-spectrometry. Purified Ribo-T was mixed in a 1:1 stoichiometric ratio with SILAC-labeled wt ribosomes containing $^{13}C/^{15}N$ Arg and Lys. Ribosomal proteins were digested with trypsin or LysC. Resulting peptides were fractionated and analyzed via LC-MS/MS using LTQ-Orbitrap XL (Thermo Scientific). Data analysis was performed using MaxQuant v1.3.0.5[52], with default settings using E. coli MG1655 protein sequence database from UniProtKB (09.09.2016). Ribosomal protein content was calculated according to "heavy"/"light" peptide ratio[53].

**Preparations of rRNA for the analysis of the ends processing**. The overnight cultures were diluted 100-fold in LB media supplemented with 100 µg/ml of ampicillin and 30 µg/ml of spectinomycin. After reaching the $A_{600} = $ ~0.5, cells were harvested by centrifugation at 5000 rpm for 10 min (4 °C) in the JA25.50 rotor (Beckman). Pellets were resuspended in ice-cold lysis buffer 2 (20 mM Tris–HCl, pH 7.5, 15 mM $MgCl_2$, 1 mg/ml lysozyme). After 5 min incubation on ice, sodium deoxycholate (Sigma) was added to the final concentrations of 0.3% (w/v) and 2 units of RQ1 DNase (Promega) were added to the buffer. After 3 min incubation in ice, cell debris was harvested by centrifugation at 20,000$g$ for 15 min (4 °C). Supernatant was loaded on the sucrose cushion (20 mM Tris–HCl, pH 7.5, 10 mM $MgCl_2$, 500 mM $NH_4Cl$, 0.5 mM EDTA, 2 mM β-mercaptoethanol, 30% (w/w) sucrose) and centrifuged for 90 min at 213,000$g$ (4 °C) in the TLS-55 rotor (Beckman). Ribo-T or ribosome pellets were resuspended in water and rRNA was isolated by phenol–chloroform extraction and ethanol precipitation.

**Analysis of rRNA ends**. Processing of the rRNA 5′ end was analyzed by primer extension. Briefly, 10 pmol of the primer 5PES (5′ ACCGTTCGACTTG-CATGTGTT 3′) was radio-labeled in a 10 µl reaction containing 1× PNK buffer A (Thermo Fisher Scientific), 15 µCi γ-[$^{32}P$] ATP (6000 Ci/mmol) (Perkin Elmer) and 10 U T4-polynucleotide kinase (Thermo Scientific). The reactions were incubated 30 min at 37 °C and the enzyme was inactivated by heating for 2 min at 95 °C. The radiolabeled primer (0.5 pmol) was annealed to 1.25 µg of rRNA in the hybridization buffer (50 mM HEPES, pH 7.0, 100 mM KCl) in the final volume of 4 µl in a PCR tube. The tube was placed in a PCR machine, heated at 99 °C for 1 min and then cooled over 15 min to room temperature. An equal volume of reverse transcription mixture [130 mM Tris–HCl, pH 8.5, 10 mM $MgCl_2$, 10 mM DTT, 187.5 µM of each dNTPs (Roche), 2 U of AMV reverse transcriptase (Roche)] was added and the reactions were incubated for 20 min at 42 °C. To reactions were stopped by addition of 120 µl of the buffer S (0.8 mM EDTA pH 8.0, 84 mM NaOAc, 70% ethanol). Tubes were incubated for 1 h at −80 °C and the reaction products were pelleted by centrifugation (30 min, 20,000$g$, 4 °C). After the removal of supernatants, pellets were dried, dissolved in the formamide buffer (80% formamide, 10 mM EDTA, pH 8.0, 1 mg/ml xylene cyanol, 1 mg/ml bromophenol blue) and loaded on the 6% denaturing polyacrylamide sequencing gels. Sequencing reactions prepared using the same 5PES primer and the 150 ng of pAM552 DNA template were loaded onto the gel alongside with the primer extension samples. After electrophoresis, the gels were dried, exposed to the phosphorimager screens and scanned on the Typhoon PhosphorImager (GE Healthcare). Images were analyzed by ImageJ software[54].

Processing of the rRNA 3′ ends was analyzed by 3′ RACE[29] using E. coli 5S rRNA as the ligation adaptor. Briefly, ~15 µg of rRNA samples were dephosphorylated by incubation in 30 µl reaction volume with shrimp alkaline phosphatase (SAP) (Thermo Fisher Scientific) following the manufacturer's protocol. Dephosphorylated RNA was purified by phenol–chloroform extraction and ethanol precipitation. E. coli 5S rRNA (Boehringer Mannheim) was used as the 3′ RACE adapter. For that, 24 µg 5S rRNA were pre-treated for 1 h at 0 °C in 100 µl of 10 mM $NaIO_4$ (in order to oxidize the 3′ ribose) and purified by the ethanol

precipitation. The oxidized 5S rRNA (2 µg) was ligated to 15 µg of the SAP-treated rRNA by an overnight incubation at 16 °C in 20 µl of 1× T4 RNA ligase buffer containing 1 mM ATP and 10 U of T4 RNA ligase (Thermo Fisher Scientific). rRNA samples ligated to the 5S rRNA adapter were purified by phenol–chloroform extraction and ethanol precipitation. The 5S rRNA-specific primer 3PES (10 pmol) (5′ GTTTCACTTCTGAGTT 3′) was annealed to the ligated RNA and the primer extension reaction was carried out as described above. Resulting cDNA was used as a template for the PCR reactions in which the 5S-specific primer R (5′ GTTTCACTTCTGAGTTCGGC 3′) and 16S-specific primer wt-F (5′ CCTGCGGTTGGATCACCTCC 3′) were used. The R primer was radiolabeled prior to the reaction according to the protocol described above. For the analysis of processing of the orthogonal Ribo-T or 16S rRNA, the PCR reactions were carried out using the R primer in combination with either the wt-F primer or the oASD-specific primer o-F (5′ GCGGTTGGATCATTGTGG 3′). All PCR reactions were carried out using AccuPrime DNA Polymerases (Thermo Fisher Scientific) under the following conditions: 96 °C, 1 min followed by 30 cycles of 94 °C, 30 s → 55 °C, 30 s → 68 °C, 30 s, followed by 68 °C, 3 min. Radiolabeled PCR products were either directly analyzed on a gel or cloned into pGEM-T vector (Promega) according to the manufacturer protocol and approximately 30 clones from each sample were sequenced.

Uncropped gels representing the results of rRNA end processing are shown in the Source Data file.

**Ribo-seq and RNA-seq**. The Ribo-seq and RNA-seq analyses were performed with the E. coli SQ171fg cells expressing Ribo-T rRNA from the pRibo-T plasmid[6]. As a control, we used the same strain transformed with the pAM552 plasmid carrying rrnB operon in which the A2058G mutation, present in the Ribo-T, was engineered in the 23S rRNA gene. Cells of each sample, grown overnight in LB medium supplemented with ampicillin (100 µg/ml) and spectinomycin (50 µg/ml), were diluted 100-fold into two conical baffled 1-L flasks containing 200 ml of LB supplemented with the same antibiotics and with 0.2% glucose. All subsequent procedures were carried out in parallel on all four samples (two Ribo-T samples and two wt controls). Cultures were grown at 37 °C with vigorous shaking. When the culture density reached $A_{600}$ of ~0.5 cells were harvested by rapid filtration, frozen in liquid nitrogen, and cryo-lysed as described[32,33]. For preparation of Ribo-seq samples, the lysates (22 $A_{260}$) were treated with ~100 U of MNase, the 70S material was isolated by sucrose gradient centrifugation following the published protocol and RNA was extracted[32,33]. For preparation of RNA-seq samples, total RNA was isolated from the same cell lysates, short RNA and rRNA were subtracted using Ambion MEGAclear kit and MICROBexpress bacterial mRNA enrichment kit, respectively and RNA was fragmented using RNA Fragmentation Reagents kit from Ambion. Ribo-seq and RNA-seq samples were size-fractionated by gel electrophoresis in 15% denaturing polyacrylamide gel. RNA fragments ranging in size from ~20 nt to ~45 nt were extracted from the gel and dephosphorylated. Four Ribo-seq RNA samples (two Ribo-T samples and two wt controls) were ligated to linkers NI-810, NI-811, NI-812, and NI813 described in ref. [55]. Each sample was ligated to its own linker. Four RNA-seq samples were ligated to the same four linkers. After ligation, the excess of the linkers was removed by treatment with 5′-deadenylase and RecJ exonuclease[55]. At this stage, all the Ribo-seq samples were pooled together and all the RNA-seq samples were pooled together. After purification using Oligo Clean & Concentrator kit (Zymo Research) the pooled Ribo-seq sample was subjected to rRNA subtraction using Ribo Zero kit (Illumina). The pooled Ribo-seq and RNA-seq samples were used as templates for reverse transcription using the primer NI-802[55]. cDNA circularization was carried out as described[55]. For generation of the dual-indexed libraries, the circularized cDNA products were used as templates for PCR reactions that were carried out using primer pair AM-i71 CAAGCAGAAGACGGCATACGAGATAACCGCGGGTGA CTGGAGTTCAGACGTGTG and AM-i51 AATGATACGGCGACCACCGAGAT CTACACAGCGCTAGACACTCTTTCCCTACACGACGCTC (for the Ribo-seq sample) or AM-i72 CAAGCAGAAGACGGCATACGAGATGGTTATAAGTGAC TGGAGTTCAGACGTGTG and AM-i52 AATGATACGGCGACCACCGAGATC TACACGATATCGAACACTCTTTCCCTACACGACGCTC (RNA-seq sample). Illumina indices are underlined. The libraries were sequenced on the NextSeq Illumina platform at the UIC DNA sequencing facility.

Ribo-seq and RNA-seq data were analyzed in the following way. The universal 3′ adapter was first removed from all reads with cutadapt[56]. Cutadapt was further used to de-multiplex samples according to the ligation linker barcodes and remove the 5′ adapter sequence. Finally, we removed the first 2 and the last 5 nucleotides from each resulting read since these were deliberately randomized and added in the library design[55]. All reads were mapped to the E. coli MG1655 genome (Genbank ID: U00096.3). Kallisto (v.0.44)[57] was used to pseudo-align RNA-seq reads to the E. coli transcriptome, resulting in TPM (transcripts per million) values for each coding sequence. Sleuth (v0.30)[58] was used to perform differential expression analysis on the likelihood-ratio test.

To analyze Ribo-seq data, and assess translation efficiency, we aligned reads from all Ribo-seq and RNA-seq samples to the E. coli genome using HISAT2 (v.2.1.0)[59] and first assigned each read to the 3′ most nucleotide. We then determined the optimal offset for the assigning the P-site codon by performing a metagene analysis and aligning the initiation peak with the start codon. We found the optimal offset to be 16 nt from the 3′ end of the read, similar to a 15 nt offset

proposed previously[60]. We created wiggle files from the resulting assignments and selected high-coverage genes for our subsequent analysis by requiring that (i) they are at least 99 nt long, (ii) at least 20% of the positions within each coding sequence have non-zero numbers of mapped reads across all four of the Ribo-seq datasets (2 biological replicates for wt and Ribo-T, each), and (iii) have an average number of reads across the coding sequence greater than 0.5. Such filtering yielded 1596 high-coverage genes.

RPKM (reads per kilobase per million mapped reads) values for each gene in both Ribo-seq and RNA-seq data were calculated to normalize for differences in read depth. Translation efficiency was then calculated by dividing the RPKM in Ribo-seq by the RPKM in RNA-seq for each gene. We tested for significant differences in translation efficiency by performing a Student's $t$-test on the translation efficiency values (2 biological replicates for wt, and 2 biological replicates for Ribo-T containing cells) for each gene. The resulting raw $p$-values were corrected via the Benjamini–Hochberg false discovery rate correction technique[61].

The 5′ metagene analysis was performed on the genes that are separated by at least 50 nt from the nearest upstream gene, and the 3′ metagene analysis was performed on genes that are not directly upstream from another gene. To perform metagene analysis, we calculated a relative ribosome density across each gene (including 50 nucleotides up- and downstream of the start and stop codons) by summing the reads across all nucleotides of the gene and dividing each position by this value such that the value at each position of each gene represents a relative ribosomal occupancy of that particular position within that particular gene (and thus facilitates comparison of genes across varied expression levels). We then calculated and plotted the mean of these values at each position for each sample within 50 nt up- and down-stream of the start and stop codons. Statistical analysis of these data was performed at each nt position relative to the start codon using a mixed linear-model with gene-specific fixed effects (using the mixedlm function from the python package statsmodels) to evaluate whether the distribution of reads from the two biological replicates in each sample was significantly different.

**Reporting summary**. Further information on experimental design is available in the Nature Research Reporting Summary linked to this article.

**Code availability**. The computational pipeline is available at https://github.com/adamhockenberry/ribo-t-sequencing.

## Data availability

The authors declare that all data supporting the findings of this study are available within the article and its supplementary information files or from the corresponding author upon reasonable request. The Ribo-seq and RNA-seq data reported in this paper have been deposited in the Gene Expression Omnibus (GEO) database under accession code GSE119454.

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

## Acknowledgements

The authors thank Amira Kefi for help with the initial bioinformatics analysis, Teresa Szal for assistance with some of the experiments, and Cedric Orelle for advice with some experimental procedures. The authors are grateful to Maxim Svetlov and Yury Polikanov for the critical reading of the manuscript. This work was supported by the NIH grant R35 GM127134 (to A.S.M.), Institutional Research Funding projects of the Estonian Ministry of Education and Research [IUT20-21] (to J.R.), the National Science Foundation (NSF) grant MCB-1716766 (to M.C.J.), the Human Frontiers Science Program RGP0015/2017 (to M.C.J.), the David and Lucile Packard Foundation (to M.C.J.), the Camille Dreyfus Teacher-Scholar Program (to M.C.J.).

## Author contributions

N.A.A., M.L., J.R. and A.S.M. conceived the study. N.A.A., M.L. and D.K. performed the experiments. N.A.A., M.L., A.J.H., M.C.J., N.V.-L., J.R. and A.S.M. analyzed the results. N.A.A., M.L., A.J.H., N.V.-L., J.R. and A.S.M. wrote the manuscript.

## Additional information

**Competing interests:** The authors declare no competing interests.

