## [Peer Review File · Nature Communications]

Reviewers' Comments:

Reviewer #1:

Remarks to the Author:

In all living cells the ribosome consists of two asymmetric subunits that assemble on mRNA to carry out the synthesis of the proteins. The only exception is a human-made T-ribosome engineered by Mankin and colleagues, who tethered the two subunits by fusing their ribosomal RNAs into a single molecule. T-ribosome was not designed just to satisfy the scientific curiosity of whether such forever united ribosome would work. The purpose was to enable mutagenetic studies of 23S rRNA using orthogonal ribosomes approach which was possible only for the 16S rRNA. The orthogonal ribosome approach is based on modifying the site of the ribosome responsible for recognition of translation initiation sites, so that such orthogonal ribosomes would predominantly translate special reporter mRNAs and would not significantly affect translation of endogenous mRNAs. However, this approach was limited only to the small subunit, since it is the small subunit that recognizes translation initiation sites. With covalently bound subunits, T-ribosome removed this limitation.

However, when endogenous ribosomes are substituted with T-ribosomes, the growth of the cells is significantly impaired. For the future studies based on the use of T-ribosomes It is imperative to understand what is the source of the reduced cell growth. Is it an impairment of the ribosome function as mRNA decoder and protein maker or is it an impairment in the process of the T-ribosome biogenesis? In this manuscript, Aleksashin et al successfully address this question by employing a large number of complementary approaches and provide strong evidence suggesting that the reduced growth is predominantly due to the impaired ribosome biogenesis. This is an impressive and important work that would be of interest not only to all ribosomologists, but many others working in the areas of Synthetic Biology since orthogonal ribosomes are often used in engineering artificial genetic codes and for incorporation of non-natural amino acids into proteins. Further, I also would like to praise the authors' presentation style. The manuscript is written clearly and is easy to follow and the figures are equally effective and aesthetically pleasing.

My only major criticism is the analysis and presentation of the ribosome profiling data. Looking at Figure 6A it is clear that some change in yhhQ (most likely its upregulation in t-ribosome cells) is the major difference in gene expression between wt and t-ribosome bearing cells. It is less clear if there are other genes whose expression is altered. The authors provide information on fold changes in ribosome footprint densities for some genes, but this is unclear whether these changes are statistically significant. Compare a 2-fold change from 5 to 10 to a 2-fold change from 500 to 1000. The former is much less likely to reflect the reality than the latter. Further, an adequate statistical analysis of ribosome profiling data cannot be done on a single set of experiments and has to be replicated in order to assess the level of stochastic noise (both due to biological and technical variations). There are several different approaches for measuring differential gene expression, z-score transformation (see Andreev et al 2015 doi: 10.1186/s13059-015-0651-z) is, perhaps, the most transparent one. The approach is implemented in Trips-Viz browser (available through Riboseq.Org) and the analysis can be carried out on-line. The lack of statistical analysis is particularly relevant to the data shown in Fig. 6b, although, under the assumption that all genes encoding ribosomal proteins are expressed at similar levels, it is unlikely that a proper data analysis would significantly alter the author's conclusions, but it may strengthen them.

Further, while I agree with the authors that yhhQ expression most likely is upregulated in t-ribosome expressing cells, there are alternative explanations that could be ruled out simply by exploring the profile of ribosome density for that gene. An alternative explanation is a strong ribosome pause in yhhQ mRNA that would result in an increase of footprint density, but not necessarily in an increase of its expression. There might be other abnormalities, so it is important to provide plots of ribosome densities for this gene.

One interesting observation that the authors' made based on metagene profiles is that the t-ribosomes are paused at the start and stop sites of protein-coding ORFs. I agree that this is the

most likely explanation, but there is an alternative explanation that should be mentioned unless the authors find a way to rule it out. The alternative explanation is that it is possible that there are stages in initiation and termination where 30S subunit is bound to mRNA, but leaves no footprints. T-ribosome may lead to the protection of mRNA during these stages. It is not necessarily that the kinetics of these processes is altered, it could be just the ability of ribosomes to protect mRNA from RNase digestion.

Overall, the presentation of ribosome profiling data is a bit sketchy. For example, it would be interesting to compare the distribution of footprint lengths obtained from different ribosomes. It is also interesting to know if there is a difference in the levels of rRNA contaminations in the data obtained from two strains. And, of course, the experiments have to be replicated at least once.

Minor comments:

For the experiments shown in Figures 4 and 5, it might be helpful to indicate the locations of PCR and sequencing primers on Fig. 4a.

Page 12. The sequence of SD is given as CCUCCU, it is actually UCCUCC

Same page: "While altering 16S rRNA ASD sequence compromised the 3' end processing (Fig. 5a, lane 4) it did not further exacerbate the already slow 3' end processing of Ribo-T (compare Fig. 4b and Fig. 5a, lane 3)." Should it be 4c instead of 4b?

This is a purely discretionary comment: Wouldn't it be helpful to refer to anti-SD as aSD instead of ASD?

There are discrepancies in the way display items are cited in the text, e.g figure panel numbers are sometimes small, sometimes in caps, supplementary items sometimes spelled out as supplementary, sometimes just an addition of S to the number. While the presentation is clear, the manuscript requires some tidying up.

Reviewer #2:

Remarks to the Author:

This follow-up study of the conjoined "Ribo-T" orthogonal ribosome characterizes its assembly and maturation in some depth, concluding that poor biogenesis likely explains the slower growth of cells expressing Ribo-T rRNA. The concept and design of the Ribo-T ribosome is clever and still quite interesting. I agree with the authors that it is amazing that this audaciously re-engineered rRNA folds and matures well enough to sustain cell viability. This study could help improve the design of orthogonal ribosomes, and this report will surely interest those who are attempting to use the Ribo-T system. On a technical level, the experiments are of high quality, and the figures look gorgeous.

I can't see any new idea developing out of this paper, however, and it is not clear it would interest anyone apart from those using the Ribo-T system. They thoroughly characterize the biogenesis pathway, showing that the Ribo-T RNA complexes are processed more slowly and lack certain modifications and ribosomal proteins. The 50S subunit appears to be more severely affected than the 30S subunit. Because every aspect of biogenesis stalls at the usual late checkpoints in 30S and 50S assembly, however, they can't pinpoint the origin of the Ribo-T defect, nor have they figured out how to fix it (which might be hard). All of the data are negative, in this respect. It's not clear that one learns anything about the normal pathway of biogenesis, either. Therefore, although their experiments are beautiful and well presented, the significance of this study is quite limited.

Reviewer #3:

Remarks to the Author:

This manuscript describes a thorough multidisciplinary analysis of the altered ribosome assembly process for the fused subunit Ribo-T ribosome. Slow growth can be explained by two main effects: slower ribosome assembly, and slower initiation of translation compared to wt ribosomes. Pulse labeling confirms slower maturation, and the intermediate particles that accumulate are defective in rRNA modifications, and have altered r-protein stoichiometry and terminal RNA processing. Ribosome profiling highlights the increased residency at the start codons, and shows altered expression levels of many genes, some of which make sense with the other observations.

Overall this is an outstanding paper that in a single opus, recapitulates 40 years of effort expended on the wt ribosome. I believe that this manuscript is both of high interest to the synthetic biology and ribosome fields, and is technically sound. Ribo-T presents an intriguing opportunity to unravel the complex interactions that occur during ribosome assembly. Both the tethering and the circular permutation must dramatically alter the order of events in wt cells, and this study nicely reveals the plasticity of the assembly process, and provides important clues to the coupling, or lack thereof, between key steps in assembly.

I strongly recommend publication.

Minor points to consider.

1) It is not immediately obvious that the pulse labeling data is consistent with a 2-fold increase in doubling time. My intuition tells me there should be a more significant accumulation of intermediates. I don't believe that more experiments need to be done, but it might be useful to quantify the area of the pulse labeling curves and to quantify the labeling rates in terms of precursor pool sizes. One thing to consider is the possibility of degradation of stalled assembly intermediates. I believe in this case, there would be reduced efficiency of chasing the label into 70S. That could reconcile the lack of major intermediates with the slow growth. The present interpretation could also be just fine, but quantitation of the existing data might resolve this.

2) The r-protein alterations for the large subunit resemble depletions that have been observed for other LSU perturbations, and it might be mentioned how the set of depletions relates to L17-depletion, the Nierhaus intermediates recently characterized by Spahn, and some of the Bacillus work from Ortega and Britton.

3) It does not appear that the pseudouridines were probed, and it might be mentioned that the methyl modifications only were probed.

4) Are there any interesting correlations in the r-protein expression levels and their membership in polycistronic operons?

Reviewer 1.

My only major criticism is the analysis and presentation of the ribosome profiling data. Looking at Figure 6A it is clear that some change in yhhQ (most likely its upregulation in T-ribosome cells) is the major difference in gene expression between wt and t-ribosome bearing cells. It is less clear if there are other genes whose expression is altered. The authors provide information on fold changes in ribosome footprint densities for some genes, but this is unclear whether these changes are statistically significant. Compare a 2-fold change from 5 to 10 to a 2-fold change from 500 to 1000. The former is much less likely to reflect the reality than the latter. Further, an adequate statistical analysis of ribosome profiling data cannot be done on a single set of experiments and has to be replicated in order to assess the level of stochastic noise (both due to biological and technical variations). There are several different approaches for measuring differential gene expression, z-score transformation (see Andreev et al 2015 doi: 10.1186/s13059-015-0651-z) is, perhaps, the most transparent one. The approach is implemented in Trips-Viz browser (available through Riboseq.Org) and the analysis can be carried out on-line. The lack of statistical analysis is particularly relevant to the data shown in Fig. 6b, although, under the assumption that all genes encoding ribosomal proteins are expressed at similar levels, it is unlikely that a proper data analysis would significantly alter the author's conclusions, but it may strengthen them.

We took to heart the reviewer's concern about the fact that ribosome profiling (Ribo-seq) was performed on a single sample. Therefore, in order to make the Ribo-seq data even more reliable and also amenable to statistical analysis, we carried a new Ribo-seq experiment which was enhanced in several ways compared to the original one. Firstly, now we collected two independent datasets (two biological replicates) for the Ribo-T and control cells. Secondly, we supplemented Ribo-seq with RNA-seq analysis performed on all four samples (two biological replicates of both Ribo-T and control cells). These data allowed us to calculate not only the absolute translation of individual genes, but also translation efficiency. Finally, as a control we now used the strain in which the ribosomes carried the A2058G mutation in the 23S rRNA gene – the same mutation that is present in Ribo-T.

We have then repeated and extended our bioinformatics analyses on the newly-obtained datasets, supplementing it with the rigorous statistical evaluation as proposed by the reviewer. Importantly, our previous conclusions remained essentially unchanged except that in the new dataset, the start- and stop codons effects (from metagene analysis) were still substantial but somewhat less pronounced than in our previous Ribo-seq experiment. Noteworthy, the extent of the increase of the start codon occupancy by Ribo-T has not changed compared to our previous

dataset that had winsorization applied. This is because the new replicated datasets were very robust and showed fewer outlying values than the previously collected dataset (which showed large differences between winsorized and non-winsorized data). As a result, winsorization does not change the look of the meta-gene plots in these updated datasets, we decided to not show this transformation in the revised manuscript.

We agree with the reviewer that statistical significance of difference in gene expression is an important parameter. Therefore, in the revised manuscript we not only highlighted the genes whose expression changes more than two fold in the Ribo-T cells but limited the list only to the genes for which p-value of the change is below 0.02 (Fig. S3a and Table S2).

Finally, for the consistency of the data presentation, we have changed the presentation of Ribo-seq data reflecting ribosomal protein expression in Figure 6. All the new data are reflected in the revised figures 6, S2 and S3 as well as in the Supplementary Table.

Further, while I agree with the authors that yhhQ expression most likely is upregulated in t-ribosome expressing cells, there are alternative explanations that could be ruled out simply by exploring the profile of ribosome density for that gene. An alternative explanation is a strong ribosome pause in yhhQ mRNA that would result in an increase of footprint density, but not necessarily in an increase of its expression. There might be other abnormalities, so it is important to provide plots of ribosome densities for this gene.

We agree with the reviewer that abnormal ribosome stalling during *yhhQ* translation could potentially result in an increased Ribo-seq score without more active expression of the full-length gene product. However, our new data, including RNA-seq show that the primary reason for the increased *yhhQ* expression is due to its up-regulated transcription (see Figs 6 and S3 in the revised manuscript). We looked at Ribo-seq coverage plots and did not see any abnormalities during translation of the *yhhQ* gene that would indicate any stalling events. Therefore, we refrained from showing the *yhhQ* Ribo-seq plots in the paper. All these data have been deposited to the GEO database and will be readily available.

Overall, the presentation of ribosome profiling data is a bit sketchy. For example, it would be interesting to compare the distribution of footprint lengths obtained from different ribosomes. It is also interesting to know if there is a difference in the levels of rRNA contaminations in the data obtained from two strains. And, of course, the experiments have to be replicated at least once.

In the revised manuscript, we replicate the results and provide the information about read length distribution in Fig. S2a,b. Additionally, the statistics of read mapping is now shown in Table S1. Since we have now implemented a modified Ribo-seq protocol we have also extended the corresponding section in Materials and Methods. Because we did not observe any specific trends in the read length distribution or stable RNA contamination in the Ribo-T samples, we saw little reason for discussing these aspects of the data in the manuscript.

Minor comments:

For the experiments shown in Figures 4 and 5, it might be helpful to indicate the locations of PCR and sequencing primers on Fig. 4a.

The sites recognized by the primers would be difficult to show in Fig. 4a because the primer used for the 5' end mapping falls within the dotted line representing Ribo-T and annealing site for the primer used for the 3' end mapping is located further downstream from the sequence shown in that figure. Instead, in the revised manuscript, we now show the position of the primer sites within the relevant sequences in Supplementary Figure S1. Specifically, we have added a new panel to this figure (panel a) showing positions of the primers used for the analysis of the rRNA 5' and 3' end maturation.

Page 12. The sequence of SD is given as CCUCCU, it is actually UCCUCC

We believe the reviewer refers to the ASD in 16S rRNA, not the SD sequence in mRNA. Being this the case, we confirm that the ASD sequence in 16S rRNA, presented in the conventional 5' to 3' orientation, is in fact CCUCCU, which is complementary to the consensus SD sequence 5'-AGGAGG-3' in mRNA.

Same page: "While altering 16S rRNA ASD sequence compromised the 3' end processing (Fig. 5a, lane 4) it did not further exacerbate the already slow 3' end processing of Ribo-T (compare Fig. 4b and Fig. 5a, lane 3)." Should it be 4c instead of 4b?

Thanks for catching this mistake. It has been corrected.

This is a purely discretionary comment: Wouldn't it be helpful to refer to anti-SD as aSD instead of ASD?

There is no convention about abbreviation of anti-SD sequence. Some authors use aSD while others prefer ASD. Because we use a low-case 'o' for 'orthogonal', we would prefer to keep ASD for anti-SD sequence in this paper.

There are discrepancies in the way display items are cited in the text, e.g. figure panel numbers are sometimes small, sometimes in caps, supplementary items sometimes spelled out as supplementary, sometimes just an addition of S to the number. While the presentation is clear, the manuscript requires some tidying up.

We have done our best to correct typos and other discrepancies in the revised version.

Reviewer 2.

I can't see any new idea developing out of this paper, however, and it is not clear it would interest anyone apart from those using the Ribo-T system. They thoroughly characterize the biogenesis pathway, showing that the Ribo-T RNA complexes are processed more slowly and lack certain modifications and ribosomal proteins. The 50S subunit appears to be more severely affected than the 30S subunit. Because every aspect of biogenesis stalls at the usual late checkpoints in 30S and 50S assembly, however, they can't pinpoint the origin of the Ribo-T defect, nor have they figured out how to fix it (which might be hard). All of the data are negative,

in this respect. It's not clear that one learns anything about the normal pathway of biogenesis, either. Therefore, although their experiments are beautiful and well presented, the significance of this study is quite limited.

We fully respect the reviewer's opinion and we accept the blame for our failure to properly highlight what we view as the significance of our study in the original manuscript. In the revised version we have tried to remedy this problem. Specifically, we emphasize the following points:

i) We believe that identifying the problems is the first and crucial step for solving them and that our work, which has demonstrated that the key Ribo-T problem is assembly, not functionality, will pave the way for its further improvement. In a broader sense, we demonstrate the conceptually-important notion that assembly should be one of the keystone considerations for the future ribosome engineering efforts. We have offered several specific ways for remedying the Ribo-T assembly problems and are currently pursuing some of the mentioned approaches ourselves.

ii) Our studies of Ribo-T assembly have illuminated important aspects of biogenesis of wt ribosome. The delayed transcription and incorporation of the 16S 3' segment, dictated by the Ribo-T architecture, revealed that RsmH and RsmI enzymes, responsible for introducing the C1402 modification, recognize not the sequence, but rather the h44 structure in the context of the small ribosomal subunit. We also present evidence for another novel concept, that the final trimming of 16S rRNA does not rely on the presence of the anti-Shine-Dalgarno sequence. We further propose the idea, based on our findings, that the evolutionary preservation of the two-subunit architecture of the ribosome might be driven in part by the necessity of introducing posttranscriptional modifications at the subunit interface. While this idea is new, it will need further studies beyond the scope of this manuscript to be fully defined.

iii) Our Ribo-seq studies of the Ribo-T cells show unexpected overexpression of *yhhQ* gene. This finding is hard to explain on the basis of our current knowledge of the ribosome function and assembly and thus could open new directions of research in this field.

Reviewer 3.

Minor points to consider.

1) It is not immediately obvious that the pulse labeling data is consistent with a 2-fold increase in doubling time. My intuition tells me there should be a more significant accumulation of intermediates. I don't believe that more experiments need to be done, but it might be useful to quantify the area of the pulse labeling curves and to quantify the labeling rates in terms of precursor pool sizes. One thing to consider is the possibility of degradation of stalled assembly intermediates. I believe in this case, there would be reduced efficiency of chasing the label into 70S. That could reconcile the lack of major intermediates with the slow growth. The present interpretation could also be just fine, but quantitation of the existing data might resolve this.

This is an excellent point. We added the quantification results to Fig. 1b and mention the implications in the revised manuscript indicating that a fraction of the 55S particles could represent the dead-end assembly products subjected eventually to abortive degradation.

2) The r-protein alterations for the large subunit resemble depletions that have been observed for other LSU perturbations, and it might be mentioned how the set of depletions

relates to L17-depletion, the Nierhaus intermediates recently characterized by Spahn, and some of the Bacillus work from Ortega and Britton.

We thank the reviewer for bring this point to our attention. In the Discussion section of the revised manuscript we discuss the relevance of the Ribo-T 55S intermediates to the assembly intermediates of the large ribosomal subunits and added the references.

3) It does not appear that the pseudouridines were probed, and it might be mentioned that the methyl modifications only were probed.

Pseudouridines have been in fact probed (they are well resolved from the four main nucleotides by HPLC). The corresponding bars are shown in Fig 2b right after the four main nucleotides. However, because the pseudouridine symbol can be easily overlooked, in the revised manuscript we have directly indicated in the text that pseudouridines have been probed.

4) Are there any interesting correlations in the r-protein expression levels and their membership in polycistronic operons?

We have not noted any particular correlation between changes in r-protein expression and their membership/position within the polycistronic operons. Therefore, we do not discuss this point in the paper.

Reviewers' Comments:

Reviewer #1:

Remarks to the Author:

I am fully satisfied with the way the authors addressed my comments. The extended ribosome profiling analysis makes the work far more solid. I have no further suggestions.